# Concept Discovery for The Interpretation of Landscape Scenicness

**Pim Arendsen [1], Diego Marcos [1,*] and Devis Tuia [1,2]**

[1]  Laboratory of Geo-information Science and Remote Sensing, Wageningen University,
    6708 PB Wageningen, The Netherlands; pim.arendsen@wur.nl (P.A.); devis.tuia@epfl.ch (D.T.)

[2]  Environmental Computational Science and Earth Observation Laboratory,
    Ecole Polytechnique Fédérale de Lausanne, 1950 Sion, Switzerland

*  Correspondence: diego.marcos@wur.nl

**Abstract:** In this paper, we study how to extract visual concepts to understand landscape scenicness. Using visual feature representations from a Convolutional Neural Network (CNN), we learn a number of Concept Activation Vectors (CAV) aligned with semantic concepts from ancillary datasets. These concepts represent objects, attributes or scene categories that describe outdoor images. We then use these CAVs to study their impact on the (crowdsourced) perception of beauty of landscapes in the United Kingdom. Finally, we deploy a technique to explore new concepts beyond those initially available in the ancillary dataset: Using a semi-supervised manifold alignment technique, we align the CNN image representation to a large set of word embeddings, therefore giving access to entire dictionaries of concepts. This allows us to obtain a list of new concept candidates to improve our understanding of the elements that contribute the most to the perception of scenicness. We do this without the need for any additional data by leveraging the commonalities in the visual and word vector spaces. Our results suggest that new and potentially useful concepts can be discovered by leveraging neighbourhood structures in the word vector spaces.

**Keywords:** interpretability; word embedding; manifold alignment

## 1. Introduction

The combination of advances in deep learning methods, in particular in the form of deep Convolutional Neural Networks (CNN), and the abundance of User Generated Content (UGC) opens up the possibility of studying how users perceive their surroundings at an unprecedented level of detail and at scale. One example is the estimation of scenicness (i.e., landscape beauty) based on outdoor images. Understanding what is perceived as beautiful in a landscape is important for the tourism industry and could be employed in recommender systems. The crowdsourcing experiment ScenicOrNot [1], where volunteers scored landscape images of the United Kingdom (Figure 1), has been used to distill the image-based perception of landscape beauty by training a deep CNN to predict scenicness [2,3]. However, CNN models are complex and hard to interpret, which prevents the understanding of which cues are being used to solve the problem: in other words, we can predict if a scene is pretty, but we still do not know why, unless we speculate *a posteriori*. This is exacerbated by the fact that perception-related tasks are subjective in nature, making it harder to validate the results.

While beauty is often seen as a subjective matter, recent research has shown that the scenicness associated to an image is often coupled with specific themes or concepts like mountains or coast [2,3]. Previous work has shown that it is even possible to reduce scenicness estimation to an objective, more interpretable task by using a *semantic bottleneck* consisting of detected concepts in the images [4]. This type of approach has the advantage of improving our control over which elements of the image

are allowed to influence the scenicness estimation. For instance, if we are interested in estimating the scenic value of a place, we would like this estimation to be independent of confunding factors such as photographer biases or specific lightning conditions. However, such an approach relies on a manually chosen set of concepts that are expected to relate to scenicness and an auxiliary dataset of images where the presence of these concepts is annotated. While providing interpretability on a set of pre-defined concepts, this approach is very rigid and does not allow for exploration of unknown concepts that may arise from the data.

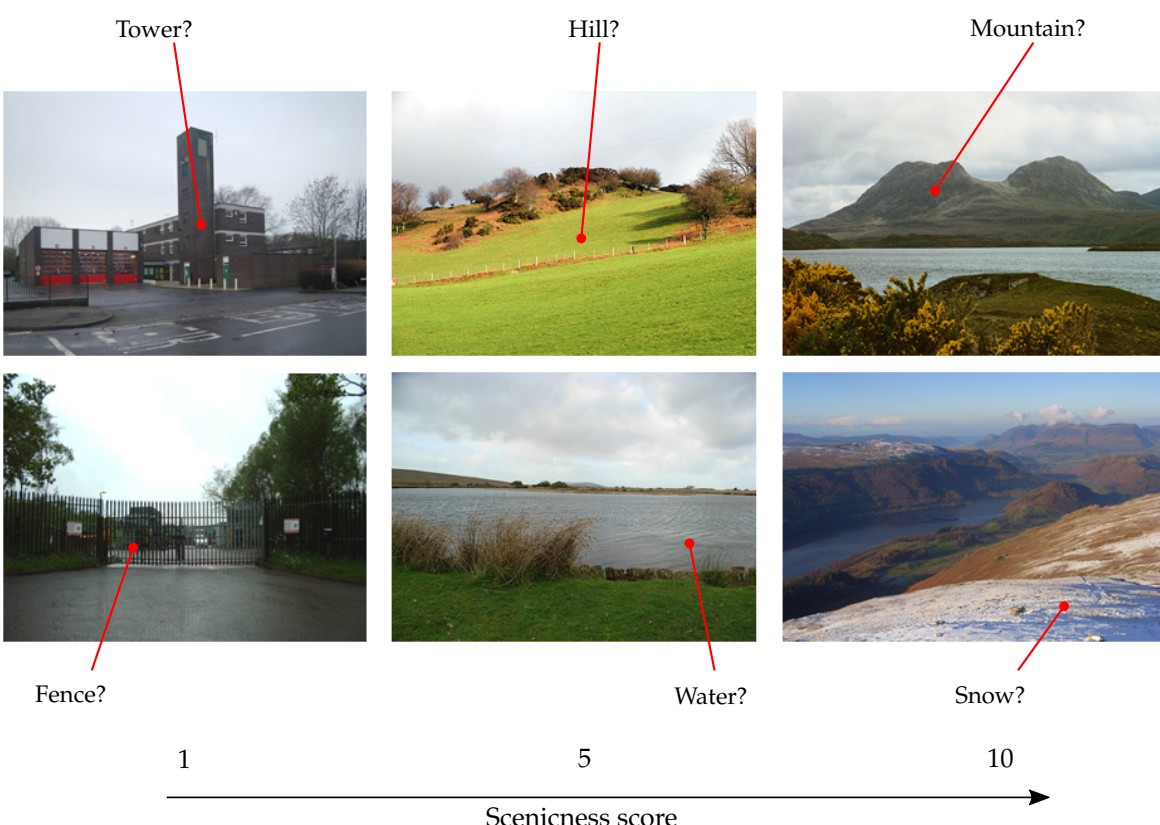

**Figure 1.** Images from the ScenicOrNot dataset, ranked from low scenicness (**left**) to high scenicness (**right**). In this work, we extract semantic concepts (water, snow, etc.) related to landscape beauty.

In this paper, we explore the potential of discovering new semantic concepts related to landscape beauty present in the data. We achieve that by expanding the set of concepts available in a visual dataset of concepts (e.g., materials, textures, objects) with a series of new concepts found by comparing visual vectors (from the CNN feature space) and word embedding vectors. This is in contrast to existing methods for concept discovery, where either new concepts are not assigned a semantic label [5], or a human-in-the-loop systems [6] or per image annotations [7] are used in order to learn the new concepts. We use manifold alignment [8] to map together the heterogeneous (different structure and different dimensionality) data manifolds of visual and word concepts, which allows to obtain hints about new concepts related to scenicness that are not captured by the initial dataset, without the need for any additional supervision.

The results on the ScenicOrNot dataset show that the proposed method can be used to explore which concepts are related to the task of scenicness estimation among those encoded by a dictionary of word embeddings.

## 2. Methodology

The proposed methodology is summarized in Figure 2. Below, we present it in three parts: the definition of the CAVs, the attribution of the CAVs to new images and the exploration of new concepts by aligning word embeddings (Code available at https://github.com/Pimmmm/Concepts_Landscape_Scenicness).

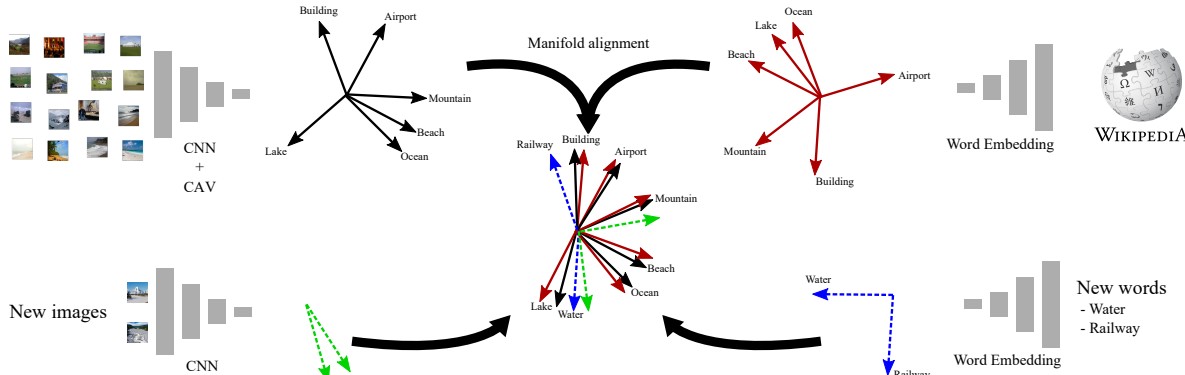

**Figure 2.** Overview of the proposed methodology. Concept Activation Vectors (CAVs) obtained using a visual dataset of concepts (top, left) are aligned with the corresponding word embeddings (top, right) using manifold alignment.

### 2.1. Concept Activation Vectors

To extract prototypical representations of semantic concepts, we use the Concept Activation Vectors (CAVs), recently introduced by [9], as a method to interpret deep CNNs without having to retrain them. CAVs use the internal high-dimensional representation of a model (typically the activation vector on a fully connected layer) to translate the model's features into human understandable concepts, ranging from simple concepts such as colours, to complex concepts like gender.

A CAV is thus a vector representing the concept as it is captured by the layer $l$ of the model. To derive a CAV, one uses an ancillary dataset containing both images representing the concept itself and counter images, i.e., images where the concept is absent. The activation vectors at layer $l$ for these images, $\mathbf{a}_l$, are extracted and a binary linear classifier is trained to discriminate the concept images from the counter images. The CAV is the vector orthogonal to the decision boundary, i.e., the vector pointing in the direction of the concept activations (see Figure 3). In the experiments, we use a ResNet50 convolutional neural network [10], pretrained on ImageNet, in combination with a large image/concept database (Broden, see Section 3). ResNet models are the standard choice when it comes to semantic feature extraction from images, for which available pre-trained models are highly optimized. We extract the image activations from the 2048-dimensional feature space of the last fully connected layer and learn CAVs for the available concepts. Next, these CAVs are linked to scenicness, as detailed in the following section.

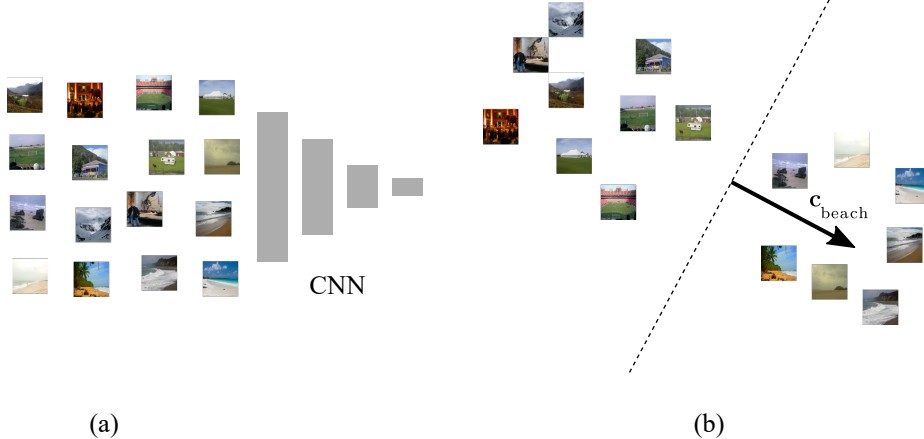

(a)        (b)

**Figure 3.** Concept Activation Vectors (CAV): Given a set of positive and negative images for a concept, in this case, beach (**a**), the activation vectors at layer $l$ are computed for each of them and a linear classifier is used to discriminate between the two sets (**b**). The CAV for concept beach is the vector orthogonal to the decision hyperplane (dashed line).

### 2.2. Linking CAVs to Scenicness

The activation vectors from all images in the dataset on scenicness (SoN, see Section 3) are extracted from layer $l$. The images from this dataset are referred to as SoN images. For each SoN image, the alignment with each CAV is calculated, which is referred to as the concept score. The concept score for image $i$ and CAV $j$, $s_{ij}$, is calculated as follows:

$$s_{ij} = \mathbf{c}_j \cdot \mathbf{a}_{il} + b_j \tag{1}$$

where $\mathbf{c}_j$ is the CAV of concept $j$, $\mathbf{a}_{il}$ is the vector of activations for image $i$ at the fully connected layer $l$ and $b_j$ is the bias term of the CAV classifier for concept $j$. The concept score provides an indication of which concepts are present in each image from SoN. The concept score is then linked to the scenicness score of the SoN images using Kendall's $\tau$ [11] correlation as in [3,4].

### 2.3. Exploring New Concepts with Manifold Alignment

To explore new concepts related to landscape beauty using CAVs would require building a large image dataset based on a prohibitively large amount of manually chosen concepts. This is unpractical, since such database would be problem-dependent and costly to build. Instead, we propose to map vectors from word embeddings into the space of CAVs. The intuition is that word embeddings are defined with large dictionaries and span a wider set of concepts than any image-based dataset, but might still have a local structure comparable to the CAVs. Since word embeddings are of different nature and dimensionality than image-based representations, we need to align the two representations to each other in a new latent vector space. To do so, we use the semi-supervised manifold alignment method [12] (Code available at https://github.com/dtuia/KEMA [13]).

Aligning the two datasets to a common representation $\mathcal{F}$ boils down to constructing two mapping functions to $\mathcal{F}$ by means of two projection matrices, $\mathbf{f}^{CAV}$ and $\mathbf{f}^{dict}$, both of dimension $(d_m \times d)$, where $d_m$ is the number of features of each dataset. The common latent space $\mathcal{F}$ is of dimension $d = d_{CAV} + d_{dict}$. Mapping to $\mathcal{F}$ requires that samples belonging to the same concept become closer, while those of different concepts are pushed far apart. Moreover, the mapping should preserve the geometry of each data manifold. Since we have terms to be minimized and others to be maximized, the problem can be solved by the following generalized eigen-decomposition:

$$\mathbf{X}((1-\mu)\mathbf{L}_g + \mu\mathbf{L}_s)\mathbf{X}^\top\boldsymbol{\varphi} = \lambda\mathbf{X}\mathbf{L}_d\mathbf{X}^\top\boldsymbol{\varphi}. \tag{2}$$

where $\mathbf{L}_g, \mathbf{L}_s$ and $\mathbf{L}_d$ are graph Laplacians, weighted by $\mu \in [0,1]$ and $\lambda \geq 0$, $\mathbf{X}$ is a block-diagonal matrix containing the data to be projected and $\varphi$ are the projection vectors aligning the data spaces. More specifically, $\mathbf{L}_g$ enforces that the geometry is preserved within every data source (i.e., neighbors in the original spaces should remain neighbors after the alignment), $\mathbf{L}_s$ promotes that vectors corresponding to the same concept across domains are mapped close together and $\mathbf{L}_d$ is a term enforcing that different concepts are mapped far away from each other. For specific matrix construction details, see [14].

The Laplacians in Equation (2) are built using CAV concepts from the visual domain and word embeddings from the text domain. Some concepts match across domains, i.e., for a CAV concept, the corresponding word embedding is present. However, to increase robustness of the latent space, both datasets include un-matched concepts, i.e., concepts for which no correspondence from the other domain is present. These concepts are used in a semi-supervised way to construct the Laplacian preserving manifold geometry, $\mathbf{L}_g$. To choose the un-matched word embeddings, we use the ten nearest neighbors to the matching concepts, while in the visual domain we use activations from single images from the final task dataset.

The eigen-problem (2) leads to the optimal projection matrix $\mathbf{F}$, which has a block structure containing the projection vectors for each domain:

$$\mathbf{F} = \left[ \sqrt{\lambda_{CAV}}\boldsymbol{\varphi}_{CAV}, \sqrt{\lambda_{dict}}\boldsymbol{\varphi}_{dict} \right] = \begin{bmatrix} \mathbf{f}_1^{CAV} & \cdots & \mathbf{f}_d^{CAV} \\ \mathbf{f}_1^{dict} & \cdots & \mathbf{f}_d^{dict} \end{bmatrix} \tag{3}$$

Once the projection matrices $\mathbf{f}^{CAV}$ and $\mathbf{f}^{dict}$ are obtained, we can project new data $\mathbf{X}_*^m$—word embeddings or new images, respectively—in the joint space by simple matrix multiplication,

$$\mathcal{P}_f(\mathbf{X}_*^m) = \mathbf{f}^{m\top}\mathbf{X}_*^m.$$

and look for concept/words distances therein (see Figure 2).

## 3. Datasets

In the experiments, we study the problem of finding semantic concepts that are relevant to the task of landscape scenicness estimation. To achieve that, we used three sources of data:

### 3.1. Landscape Scenicness

ScenicOrNot (SoN) [1] is a user generated outdoor image dataset of Great Britain, with every image rated on its scenicness-score (aesthetic value). These outdoor images are obtained directly from the Geograph project [15], where users are encouraged to provide images covering the whole surface of the British isles. A subset of this collection was then selected and shown to volunteers online to obtain the scenicness ratings.The score ranges from 1 (not scenic) to 10 (very scenic), examples are shown in Figure 1. The SoN dataset contains 212,000 images, which have been rated by at least three volunteers. For all images the longitude, latitude, number of votes, and voted scores are available. Refer to [3] for more details on this dataset.

### 3.2. Semantic Concepts

The Broadly and Densely Labeled Dataset (Broden) [16] is a combination of several image datasets: ADE [17], Open Surfaces [18], Pascal-Context [19], Pascal-Part [20] and Describable Textures Dataset [21]. The original data contains 63,000 images covering 1197 concepts. Some images are labeled at the pixel level and can be labeled with multiple concepts. Some examples are shown in Figure 4.

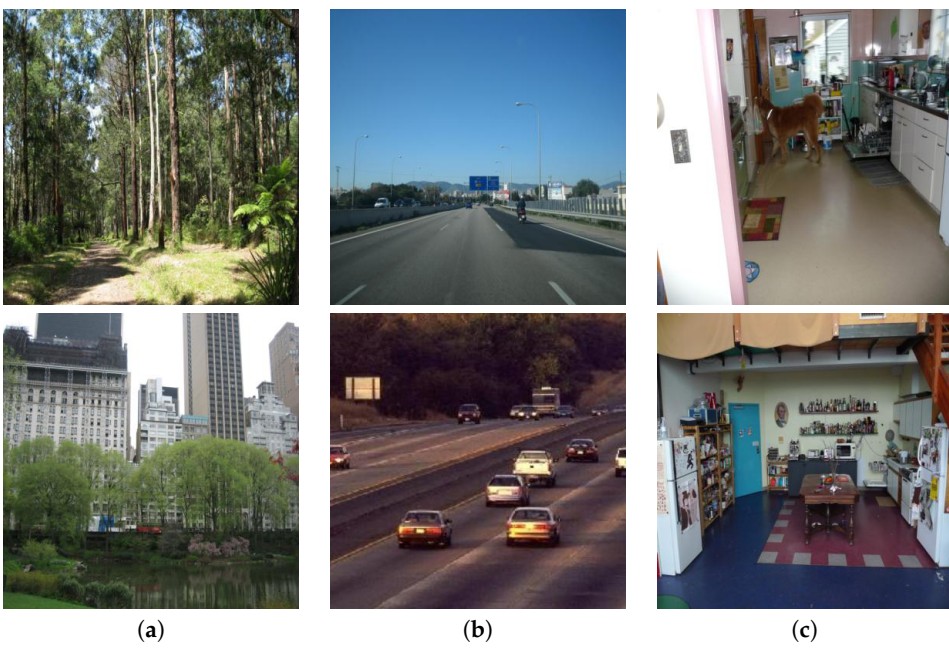

| (a) | (b) | (c) |

**Figure 4.** Example images from Broden, concepts: (**a**) Tree, (**b**) highway and (**c**) linoleum.

### 3.3. Word Embeddings

The Global Vectors (GloVe) dataset [22] contains a large dictionary of words and their corresponding word embedding. The GloVe feature space has linear substructures that capture the semantic relationships between words. The representations are learned by a unsupervised learning algorithm trained on word-to-word co-occurence statistics in a large text corpus. We used a version of the GloVe dataset containing 400,000 words, represented in a 300 dimensional space.

## 4. Results and Discussion

In this section, we present the results obtained in the ScenicOrNot dataset. We follow the same three-steps structure of the Methodology section above.

### 4.1. Deriving CAVs from Broden

In the first step, we derive CAV representations from the Broden dataset. We first applied a filter by removing labels covering less than 3% of the image to improve the image labelling, which left us with $1'091$ concepts in Broden. The data was randomly split in 70% training and 30% validation sets.

The first set of experiments was aimed at understanding the minimal amounts of concept images $a$ and counter images (in which the concept is absent) $b$ required to define the CAV appropriately. Both concept and counter images are randomly sampled from the training images in Broden. First, $a$ was determined by fixing $b = 500$ and varying $a$ in the range $a \in (1 - 100)$. If a concept did not have a hundred concept images, the maximum available number of concept images was used for defining the upper limit. For each $(a, 500)$ pair, the accuracy of the derived CAV was calculated on the 30% held out set. The procedure was repeated ten times for robustness. The top row of Figure 5 shows the accuracy of concepts highway, mountain snowy and coast with a varying number of concept images. The best $a$ is chosen to be 40, which is when the accuracy tends to plateau for the majority of concepts.

The determined $a = 40$ was used to find a preferred $b$ using the same procedure, where $b$ is incrementally sampled from $b \in (1 - 500)$. The accuracy of the same two concepts with a varying number of counter images is shown in Figure 5. The general trend in different concepts showed a stable accuracy around 200 counter images. Using the preferred $a = 40$ and $b = 200$, CAVs were derived for each concept in Broden. All CAVs with an accuracy lower than 75% were removed, as it was assumed that these CAVs were inaccurate representations of a concept, leaving us a total of 628 CAVs

for further analysis. Figure 6 shows the eight closest aligned test images with the CAVs for mountain, ship and horse.

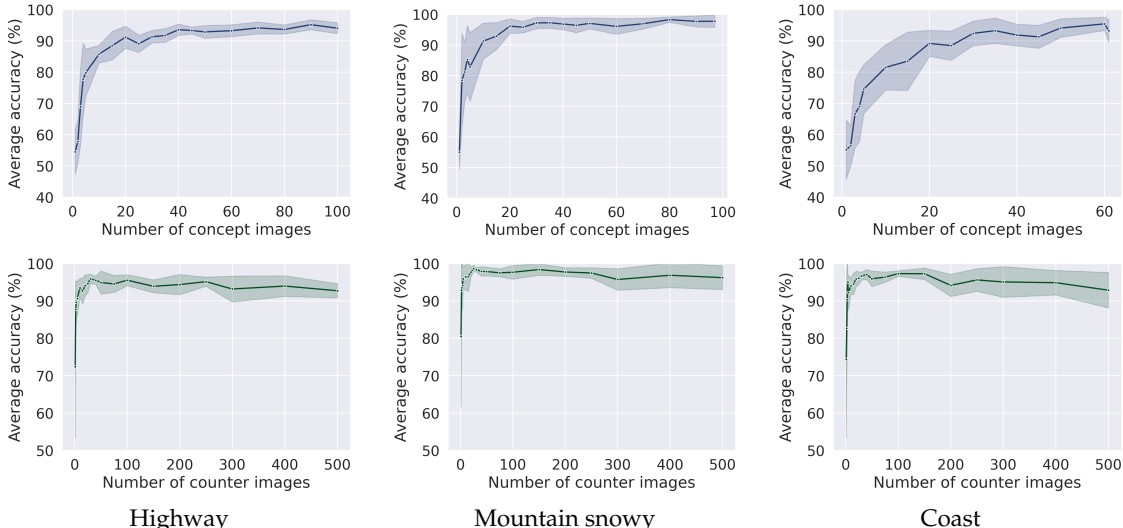

**Highway**        **Mountain snowy**        **Coast**

**Figure 5.** CAV average accuracy of concepts highway, mountain snowy and coast with a varying number of concept images (top row) and counter images (bottom row). The shadowed band indicates the standard deviation over the ten runs.

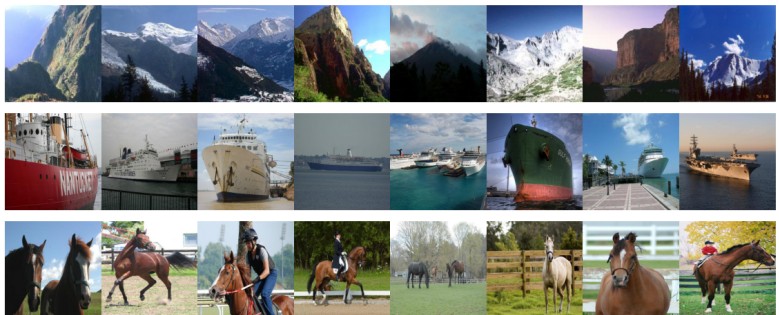

**Figure 6.** Eight nearest neighboring images in the test data to the determined Broden CAVs for concepts mountain, ship and horse.

### 4.2. Linking CAV Concepts to Scenicness

After extracting the CAVs from Broden, we calculated the concept scores $s_{ij}$ for each image $i$ in the SoN dataset with respect to each concept $j$.

The concept score was linked with the scenicness score of the SoN images, using Kendall's $\tau$ correlation coefficient. The coefficient indicates how each concept is correlated with the scenicness score, i.e., it provides an indication of concepts related to higher or lower scenicness. Table 1 presents the ten concepts showing the highest positive correlation with scenicness; these concepts relate to nature and outdoors concepts, in line with the observations of [2–4]. Similarly, Table 1 presents the ten concepts with strongest negative correlation, and indicates that man-made structures are mostly associated with unscenic landscapes.

**Table 1.** Top 10 Broden concepts positively (left) and negatively (right) correlated to scenicness.

| Rank | Concept | Correlation | Rank | Concept | Correlation |
|------|---------|-------------|------|---------|-------------|
| 1 | Canyon | 0.47 | 1 | Building | −0.39 |
| 2 | Cliff | 0.43 | 2 | Street | −0.37 |
| 3 | Island | 0.41 | 3 | Sidewalk | −0.37 |
| 4 | Valley (scene) | 0.41 | 4 | Crosswalk | −0.36 |
| 5 | Ocean | 0.40 | 5 | Parking lot | −0.35 |
| 6 | Wave | 0.40 | 6 | Windows | −0.33 |
| 7 | Mountain | 0.40 | 7 | Parking garage indoor | −0.32 |
| 8 | Valley | 0.40 | 8 | Bleachers outdoor | −0.31 |
| 9 | Smeared | 0.39 | 9 | Platform | −0.30 |
| 10 | Waterfall-block | 0.39 | 10 | Road | −0.30 |

We used the location of the SoN images to validate the concept scores obtained with clearly identifiable physical features. We selected the concept mountain and mapped the score geographically in Great Britain, as depicted in Figure 7. The area highlighted in Figure 7a gives a clear indication in which region of the UK the SoN images contain mountains. This corresponds with the elevation map of the same region, shown in Figure 7b.

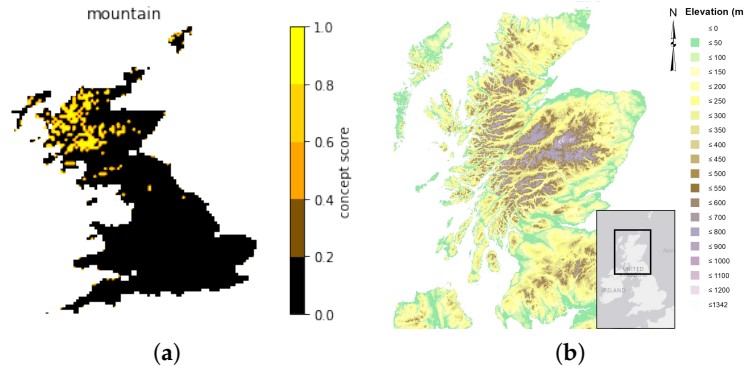

(**a**)                                                                 (**b**)

**Figure 7.** (**a**) Mountain concept score; (**b**) Elevation map of the northern UK.

### 4.3. Discovering New Concepts with Word Embeddings

To discover new concepts related to scenicness without requiring an image dataset, we aligned the manifolds of the CAV domain with the GloVe domain in a latent space. This allowed us to explore a large range of new concepts.

To build the graph Laplacians required in Equation (2), we used the concepts which have a corresponding representation in both the CAV and GloVe domain: from the initial 628 CAVs, only 302 had matching representation in both domains. However, to increase robustness of the latent space, additional non-corresponding concepts were added:

- In the CAV domain, both the non-corresponding CAVs and a random sample of SoN image vector representations were used, resulting in 5052 unmatched samples.
- In the GloVe domain, the ten nearest neighbours for each of the corresponding concepts were added, resulting in a total of 2548 samples.

The dimensionality of the CAV domain, 2048, was reduced to 100 by means of PCA. For the manifold alignment, the generalized eigen-decomposition method described in [13] requires the setting of two hyperparameters: $\mu \in [0,1]$, which trades off the alignment of corresponding concepts versus the preservation of the original manifolds' topology, and $\lambda \geq 0$, that sets the weight of the dissimilarity term. We set those parameters experimentally to $\mu = 0.9$ and $\lambda = 0.5$.

After the transformation to the latent space, the SoN images with an average scenicness score lower than 2 (unscenic) and higher than 7 (scenic) were mapped to the latent space. To explore new

concepts related to scenicness, we again used Kendall's $\tau$ to get an understanding of the relation between scenicness score and the transformed non-corresponding GloVe concepts, i.e., concepts that do not appear in the Broden dataset. Table 2 shows the 10 new concepts with the largest positive and negative correlation. In which the '*training neighbor*' column indicates the most similar concept from Broden.

To get insights on the semantics captured by the alignment in the latent space, several concepts are visualised in Figures 8 and 9. The figures show SoN and Broden images for the concept of interest with the highest concept score computed in the latent space. While the interpretation of the images is subject to interpretation bias, the visual examples in Figure 8 suggest that the new concept representations in the latent space capture the right semantics. For example, the concept outcrop mainly shows rock formations near coastal areas, thatched shows buildings with thatched roofs and arctic shows snowy environments including water (its training concept was ocean) but the captured semantics differ from those shown by snow. These results suggest that the new concepts, for which no visual ground truth is available, are able to capture the correct visual cues by leveraging their connection to other concepts.

In order to investigate some cases more in detail, we have chosen four Broden concepts and show the images in Broden and SoN that are the most aligned with them and with their GloVe neighbours. In particular, we display the four GloVe neighbours, among the total of 10 used, that obtain the highest average concept score for the top 5% highest scoring images. The first row is composed by images aligned with the training concept in the latent space and the following rows by images aligned with neighboring concepts from GloVe.

**Table 2.** Top 10 new concepts from GloVe positively (left) and negatively (right) correlated with scenicness in the latent space.

| Rank | Concept | Training Neighbor | Correlation | Rank | Concept | Training | Correlation |
|------|---------|-------------------|-------------|------|---------|----------|-------------|
| 1 | outcrop | islet | 0.54 | 1 | refrigerated | refrigerator | −0.52 |
| 2 | archipelago | island | 0.54 | 2 | expressway | highway | −0.51 |
| 3 | uninhabited | islet | 0.53 | 3 | supported | bush | −0.51 |
| 4 | wilderness | forest | 0.52 | 4 | brakes | wheel | −0.50 |
| 5 | rocky | mountain | 0.52 | 5 | concourse | mezzanine | −0.50 |
| 6 | foothills | mountain | 0.52 | 6 | closed | shed | −0.49 |
| 7 | arctic | ocean | 0.51 | 7 | profits | net | −0.48 |
| 8 | bass | guitar | 0.50 | 8 | undies | bedclothes | −0.48 |
| 9 | rugged | mountain | 0.50 | 9 | console | dashboard | −0.48 |
| 10 | unpopulated | islet | 0.50 | 10 | plastered | poster | −0.48 |

For several concepts the subtle semantics seem to be captured in the latent space which can provide hints to possible new concepts relevant to scenicness. For example, Figure 10 illustrates visual examples of SoN and Broden images related to the newly discovered concepts that are close to the training concept cottage. The new concepts thatched and ranch show an increased scenicness correlation and their corresponding SoN and Broden images seem to capture the right semantics. Figure 11 provides similar illustrations for ocean. The new concept coastline seems represented by matching images and even for coral, a concept which is not present in SoN, the most aligned images portray beach and rocky reef scenes reminiscent of environments where coral reefs do occur. In Figure 12 it is noteworthy that foothills and mountains show similar Broden images, while the SoN images tend to show gentler hills for foothills and both new concepts have a relatively high positive correlation in the latent space. The concept driveway, shown in Figure 13, has a negative correlation with scenicness in the latent space. Interestingly, its neighboring concept gravel seems to be positive related to scenicness.

ScenicOrNot images ↓ Broden images ↓

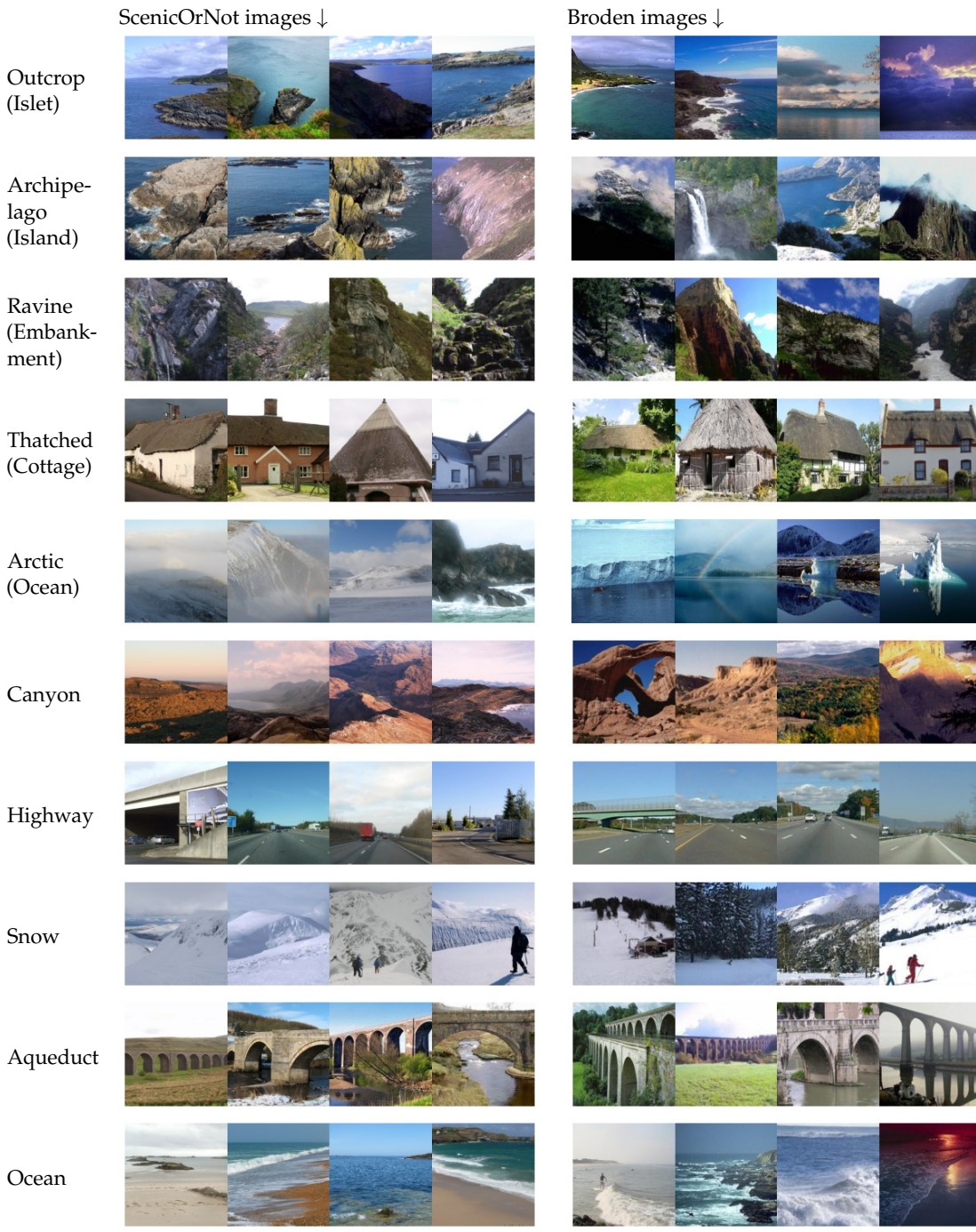

**Figure 8.** Examples of concepts for which the alignment resulted in clear image representations. For each concept, the left images show the 4 SoN images with the highest concept score in the latent space and the images of the right show the 4 Broden images with the highest corresponding concept score in the latent space. The first five concepts are from GloVe and their training neighbor is shown between parenthesis, the bottom five concepts are training neighbors.

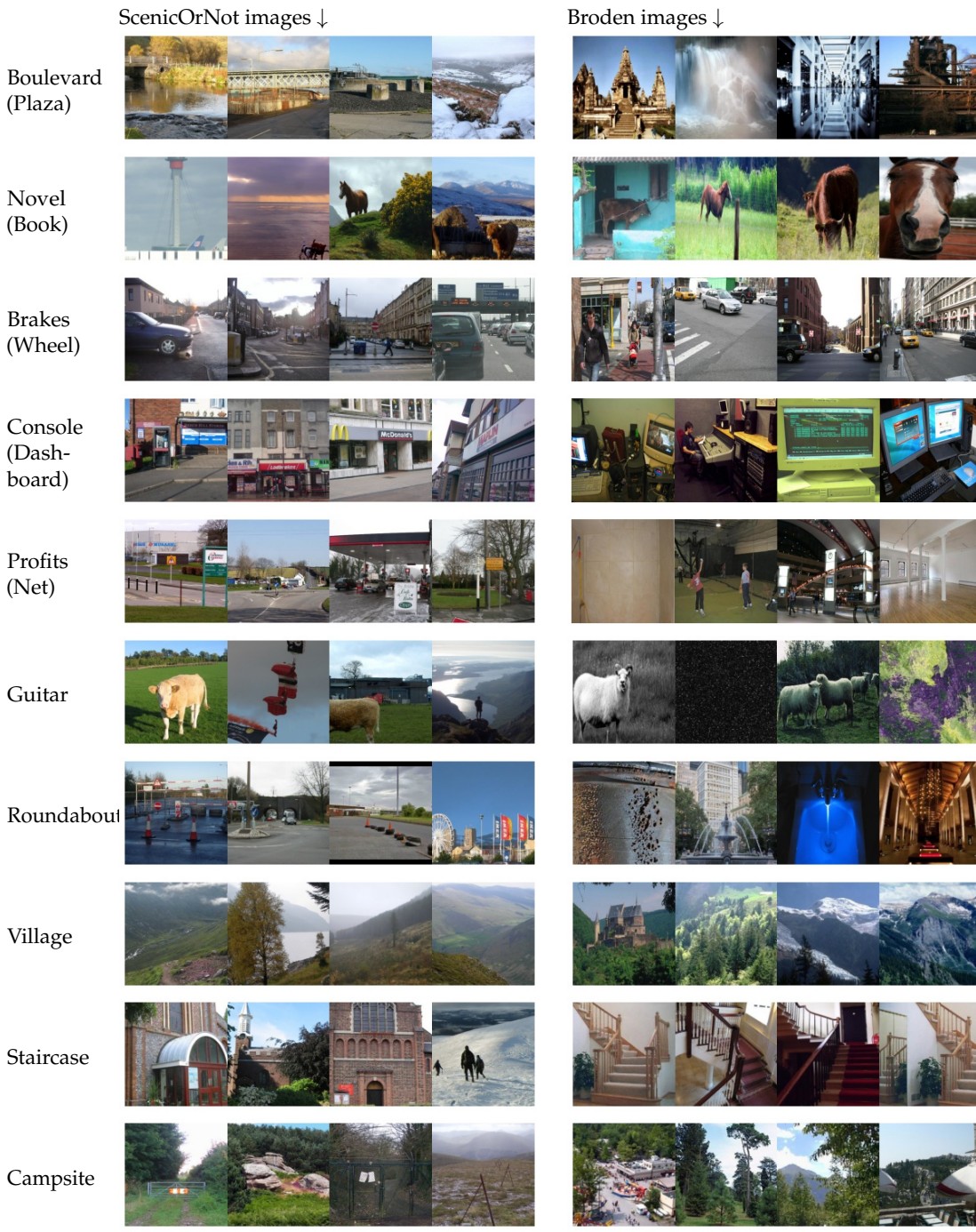

**Figure 9.** Examples of concepts for which the alignment resulted in inaccurate image representations. For each concept, the left images show the 4 SoN images with the highest concept score in the latent space and the images of the right show the 4 Broden images with the highest corresponding concept score in the latent space. Each concept's training neighbor is shown between parenthesis.

ScenicOrNot images ↓  Broden images ↓

Cottage
(Broden)
(0.081)

Bungalow
(-0.341)

Thatched
(0.202)

Farmhouse
(0.061)

Ranch
(0.283)

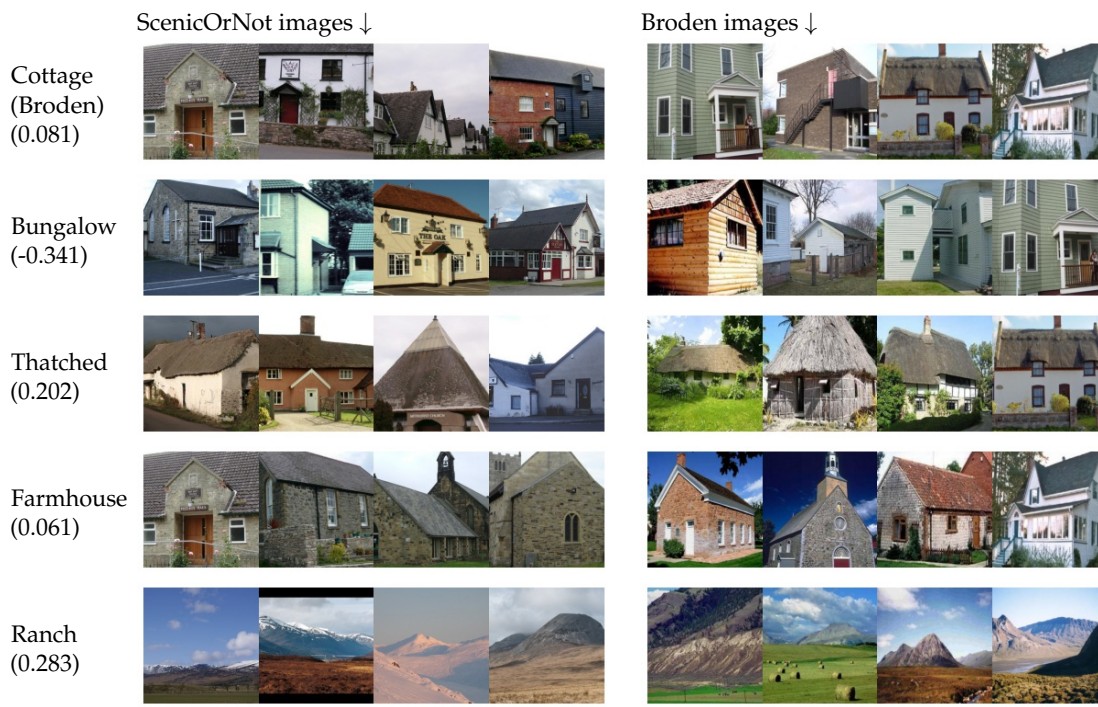

**Figure 10.** Closest images for cottage, a concept that has been trained with visual examples from Broden (top row) and its corresponding neighboring concepts from GloVe. Each concept shows its correlation with scenicness in the latent space.

ScenicOrNot images ↓  Broden images ↓

Ocean
(Broden)
(0.479)

Coastline
(0.455)

Coral
(0.494)

Islands
(0.470)

Arctic
(0.507)

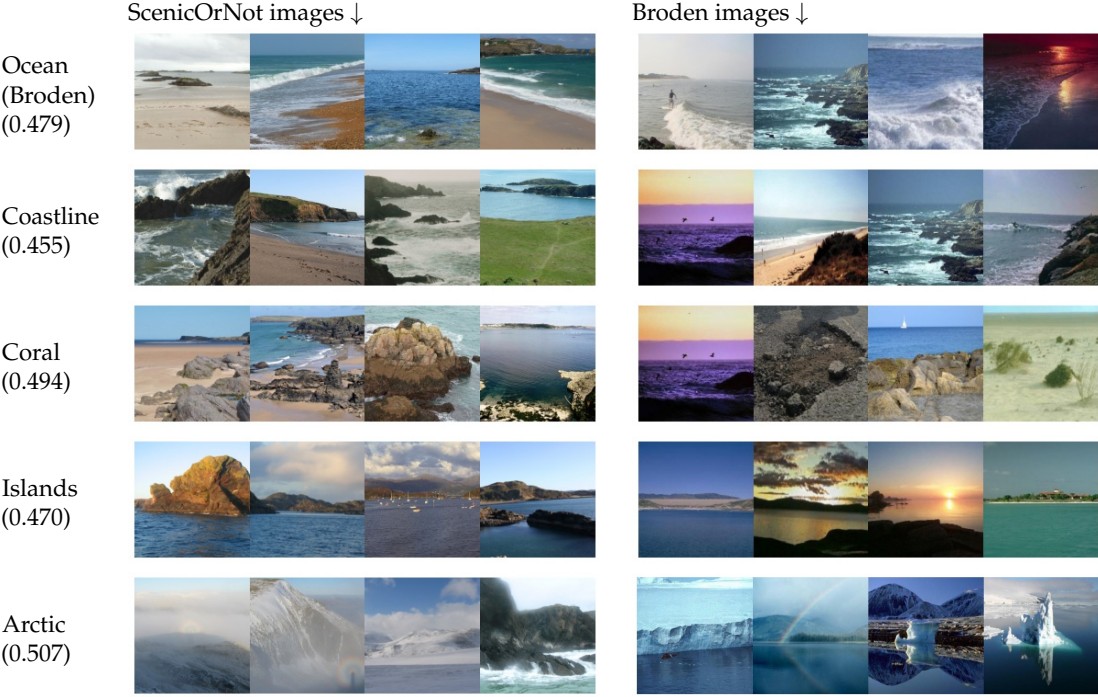

**Figure 11.** Closest images for ocean, a concept that has been trained with visual examples from Broden (top row) and its corresponding neighboring concepts from GloVe. Each concept shows its correlation with scenicness in the latent space.

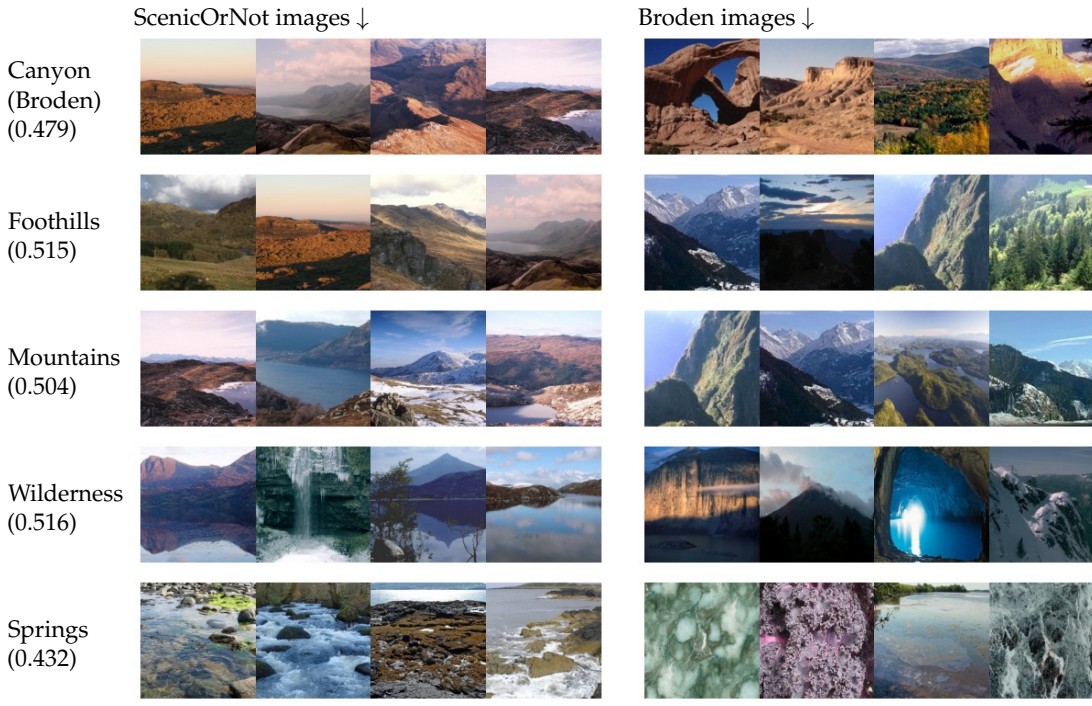

**Figure 12.** Closest images for canyon, a concept that has been trained with visual examples from Broden (top row) and its corresponding neighboring concepts from GloVe. Each concept shows its correlation with scenicness in the latent space.

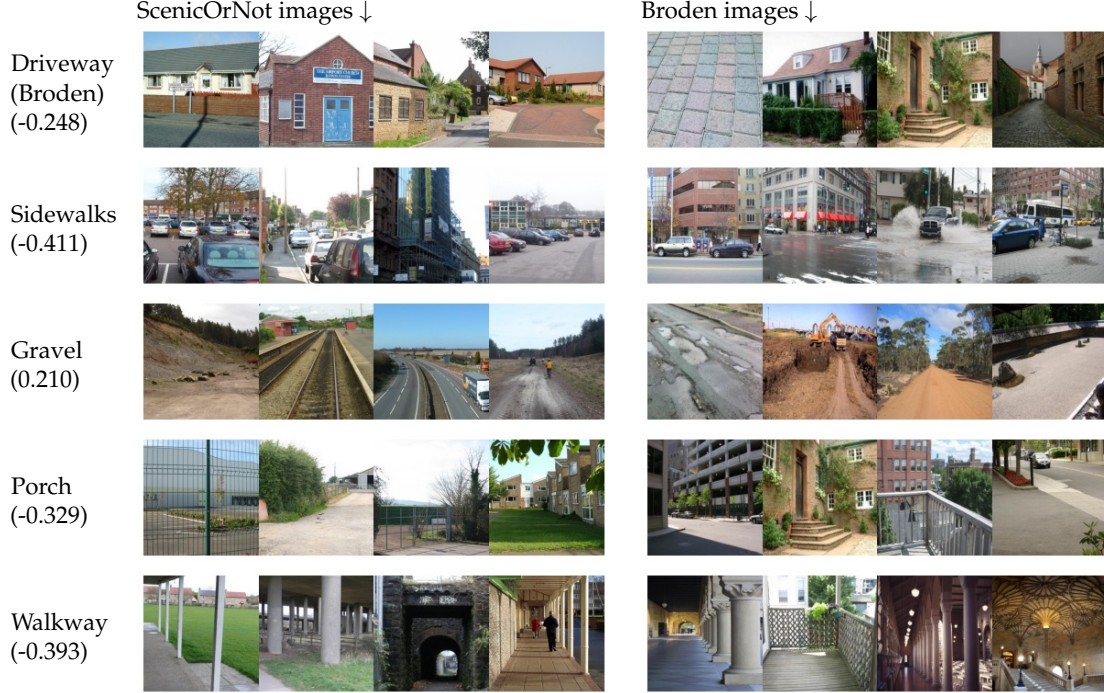

**Figure 13.** Closest images for driveway, a concept that has been trained with visual examples from Broden (top row) and its corresponding neighboring concepts from GloVe. Each concept shows its correlation with scenicness in the latent space.

### 4.4. Main Limitations of the Approach

Figure 9 suggests that new concepts need to be carefully inspected before being added to the visual dataset. We found that some of the new concepts do not capture the expected visual patterns.

Most of the inaccurate image representations can be explained by two limitations: (1) the absence of the visual concept in the SoN images or (2) a mismatch between the meaning of the visual Broden concepts and the corresponding GloVe embeddings, for instance due to polysemia.

The first limitation becomes evident when evaluating indoor-related concepts. The SoN dataset contains exclusively outdoor scenes, while Broden contains both indoor and outdoor images. As such, SoN does not have images containing most indoor concepts. For example, console in Figure 9 seems to capture reasonable semantics, as suggested by the most active images in Broden. However, due to the lack of indoor images in SoN, it is mostly activated by shopfronts, arguably the most visually similar elements to consoles in the SoN dataset, resulting in the concept being negatively correlated with scenicness. This also shows for staircase, the Broden images show staircases while SoN images seem mostly building-related. Brakes is a good example of a specific concept for which both Broden and SoN do not have clear image representations, but the images do suggest that the semantics are related to streets and cars. Thus, the concept is not necessarily inaccurate, the image datasets only do not contain images to clearly visualise the concept.

The second limitation is a 'translation error' between the visual Broden concept and its corresponding word embedding. This introduces unexpected concepts to be related to scenicness and it also contaminates concept representations in the latent space. The translation error is caused by concepts having multiple definitions, for example net (in Figure 9), which was related to, e.g., a volleyball net in Broden, in GloVe was related to money and business thus introducing profits as a new concept. Table 3 shows several concepts with polysemia in which the GloVe neighbours are related to a different meaning than the one assumed in Broden. For instance, the visual concept rock is related to geology in Broden, and still is after the alignment, as seen in Figure 14, while in GloVe rock is related the music genre, introducing music related concepts. The images clearly show that the neighboring concepts are unrelated to the initial semantics of the Broden concepts. The concept bush, although related to small trees in Broden, is pulled away from this meaning after the alignment. The images most aligned with this concept and several of their neighboring concepts are visualised in and Figure 15. Besides the mismatch in meaning resulting in unexpected new concepts, this also reduces the quality of the alignment, resulting in an invalid bush visual concept even if it is originally present in Broden.

**Table 3.** Concepts can show a mismatch in meaning between the visual concept in Broden and its corresponding embedding in GloVe. The top row concept is from Broden, the other concepts are neighbors from GloVe. Bush is related to vegetation in Broden while in GloVe it is related to the American president, net is related to, e.g., a volleybal net while in GloVe it is related to money and business, rock is related to geology in Broden while in GloVe it is related to music and coach is related to transportation in Broden while in GloVe it is related to sports.

| Bush | Net | Rock | Coach |
|---|---|---|---|
| gore | profit | band | coached |
| w. | quarter | punk | coaches |
| administration | profits | pop | coaching |
| republicans | earnings | bands | team |
| aides | income | album | football |
| democrats | revenue | rocks | basketball |
| dole | revenues | music | assistant |
| president | drop | singer | manager |
| presidential | billion | albums | players |
| republican | pretax | songs | teammates |

ScenicOrNot images ↓ Broden images ↓

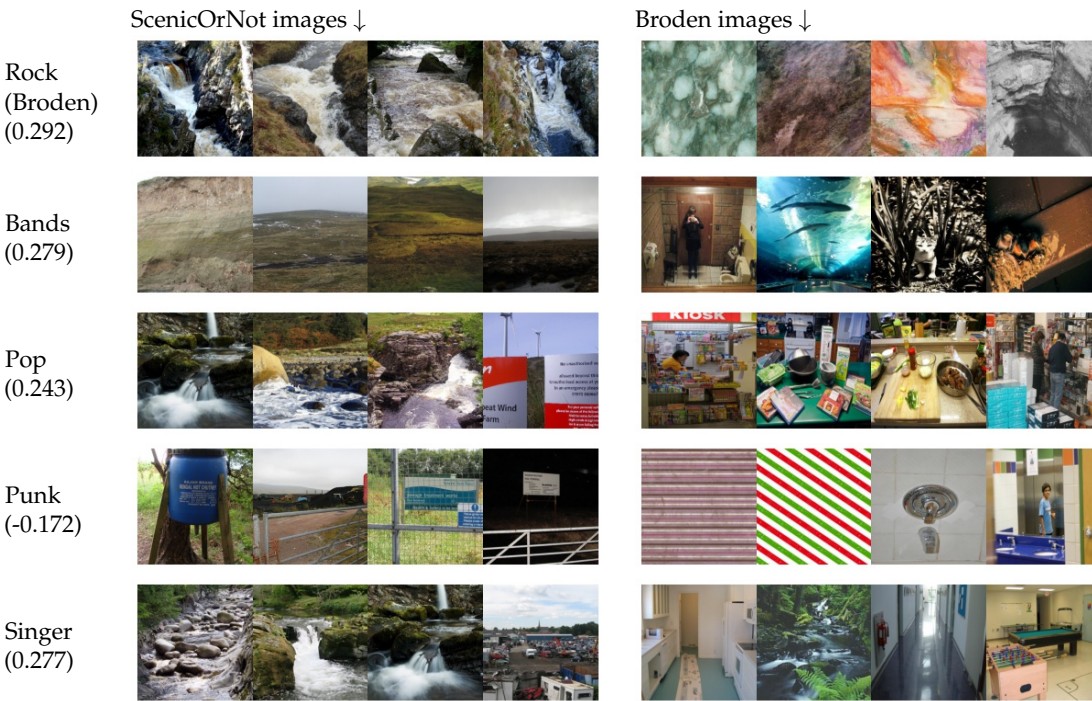

Rock
(Broden)
(0.292)

Bands
(0.279)

Pop
(0.243)

Punk
(-0.172)

Singer
(0.277)

**Figure 14.** Closest images for rock, a concept that has been trained with visual examples from Broden (top row) and its corresponding neighboring concepts from GloVe. Each concept shows its correlation with scenicness in the latent space.

ScenicOrNot images ↓ Broden images ↓

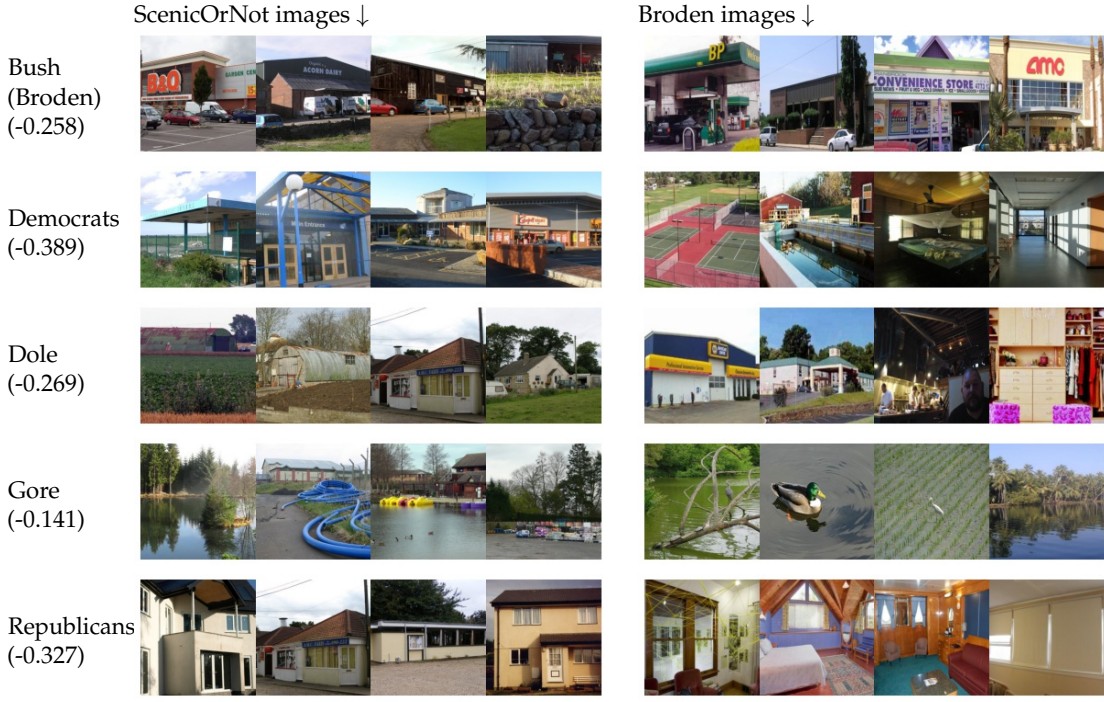

Bush
(Broden)
(-0.258)

Democrats
(-0.389)

Dole
(-0.269)

Gore
(-0.141)

Republicans
(-0.327)

**Figure 15.** Closest images for bush, a concept that has been trained with visual examples from Broden (top row) and its corresponding neighboring concepts from GloVe. Each concept shows its correlation with scenicness in the latent space.

## 5. Conclusions

We propose a methodology to explore a broad range of concepts that relate to landscape scenicness by using Concept Activation Vectors (CAVs) computed from a visual dataset of generic concepts

(the Broden image dataset). Furthermore, we explore the use of cross-domain manifold alignment to enlarge the concepts space with a corpus made of word embeddings.

Our results show that, using relatively few images, it is possible to determine a CAV that generalizes well and is able to detect the concept on new images. On average, using 40 images representing the concept already results in the best attainable accuracy, which was 85–95% for the majority of the tested concepts. Next, by linking the CAVs to the SoN scenicness we showed that, in line with previous research, nature related concepts contribute to higher scenicness in landscapes, while man-made related concepts tend to contribute to unscenic landscapes. Finally, we expanded the set of CAVs by aligning the Broden visual CAVs to their corresponding word embeddings in the GloVe domain via cross-domain manifold alignment method. Through this alignment, it became possible to search the GloVe space to find new interesting concepts for scenicness.

Our results have shown the potential as well as the limitations of aligning a visual domain with a text domain in a latent space. While not all concepts from the GloVe dataset can be linked to scenicness through this alignment, our results do suggest that the alignment is able to capture semantics which can provide new concepts without any training visual examples. At the same time, we have found that the polysemous nature of the word embedding representations are one of the main hurdles to overcome in order to ensure the usefulness of the proposed methodology. This leads us to believe that explicitly accounting for polysemy in word embeddings would be an interesting direction for future research.

**Author Contributions:** All authors have read and agreed to the published version of the manuscript. Conceptualization, D.M. and D.T.; methodology, D.M., D.T. and P.A.; experiments, P.A.; writing, P.A., D.M. and D.T.; visualization, P.A.; supervision, D.M. and D.T.

**Funding:** This research received no external funding.

**Conflicts of Interest:** The authors declare no conflict of interest.

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
