# Peer review of "Concept Discovery for The Interpretation of Landscape Scenicness"

_make, doi:10.3390/make2040022_

Round 1

Reviewer 1 Report

The paper proposes to extract visual concepts using Convolutional Neural Network (CNN). The approach uses a semi-supervised manifold alignment technique applied on a number of Concept Activation Vectors (CAVs) aligned with semantic concepts from ancillary datasets in order to obtain a list of new concept candidates to represent the perception of scenicness. 

The paper shows interesting results on standard databases. But, the work shows the following weaknesses;

1) The methods are well-known. Please, add more comments about the innovation of this proposal.

2) Many important references on concept discovery are missing. Please, upload the references and the state-of-the-art.

3) There is no doubt that the framework is able to extract visual concepts. However, the article omits details, and does not address sparsity problem nor some technical points.

A significant step consist to select proper neighborhood and exploit suitable correlation coefficients for prediction, and multiple weighting techniques have been used to enhance the performance.

When the user'correlations are computed in latent space, there should be some explanation on how are exploited the correlation coefficients in the prediction step ?

4) The experimental methodology is not clear.

My main concern with the paper is that the authors do not elaborate enough on a novelty rationale for their method compared to (their) recent work. This work must be improved and authors should explain deeper their decisions and possible extensions (or not) of the method. The authors must present comparable results from recent techniques.

Author Response

The paper proposes to extract visual concepts using Convolutional Neural Network (CNN). The approach uses a semi-supervised manifold alignment technique applied on a number of Concept Activation Vectors (CAVs) aligned with semantic concepts from ancillary datasets in order to obtain a list of new concept candidates to represent the perception of scenicness. 

The paper shows interesting results on standard databases. But, the work shows the following weaknesses;

1) The methods are well-known. Please, add more comments about the innovation of this proposal.

We would like to start thanking the reviewer for the comments and the time invested in our paper. The individual methods we have used for building our approach (CNN, GloVe, manifold alignment) are indeed well established. However, we believe that the scientific novelty of the paper is the use of these methods for the unsupervised discovery of concepts that are relevant to the particular visual task at hand (in our case, that of scenicness estimation). No previous works analyse such drivers in an unsupervised way and our proposition shows a reproducible methodology to explore out-of-domain concept spaces and actually learn new attributes relevant to the task. 

2) Many important references on concept discovery are missing. Please, upload the references and the state-of-the-art.

We agree with the reviewer that there is a connection between the proposed method and existing works on concept discovery. However, existing methods work with a different set of assumptions. For instance, [A] focuses on active learning with humans in the loop for discovering new concepts, while [B] makes use of ad-hoc image descriptions. [C] frames concept discovery as an unsupervised method, thus providing correspondences across images with the same subject, but without naming the discovered concepts. Our method leverages the commonalities between the structures of a work embedding and a CNN feature space to propose correspondences between new visual concepts and their corresponding words. We have added a section discussing these works in the introduction. We thank the reviewer for pointing this out.

[A] Wigness at al., Selectively Guiding Visual Concept Discovery, WACV14

[B] Sun et al., Automatic Concept Discovery from Parallel Text and Visual Corpora, ICCV15

[C] Collins at al., Deep Feature Factorization For Concept Discovery, ECCV18

3) There is no doubt that the framework is able to extract visual concepts. However, the article omits details, and does not address sparsity problem nor some technical points.

We appreciate the comment on the capability of the proposed approach. We have strived to detail the method as much as possible by extending section 2.1 and section 2.2., which now better clarify the CNN implementation. We also removed ambiguities on the data sources. We have also added a link to the functional code where all additional details can be inspected.

A significant step consist to select proper neighborhood and exploit suitable correlation coefficients for prediction, and multiple weighting techniques have been used to enhance the performance. When the user'correlations are computed in latent space, there should be some explanation on how are exploited the correlation coefficients in the prediction step ?

We would like to note that the scenicness prediction is unaffected by the concept discovery, and thus we have not aimed at improving the performance. The method is meant to explore potential concepts, semantically close to the ones available at training time, that explain some aspects of the scenicness prediction without modifying it.

4) The experimental methodology is not clear.

We have improved the section presenting the experimental setup (Methodology chapter), in particular by explaining the extraction of the CAVs and their link to scenicness in a clearer way. We hope this clarifies our approaches and enhances the reproducibility of the study. Also, please note that we have made the code publicly available.

My main concern with the paper is that the authors do not elaborate enough on a novelty rationale for their method compared to (their) recent work. This work must be improved and authors should explain deeper their decisions and possible extensions (or not) of the method. The authors must present comparable results from recent techniques.

The elaboration of the methodology has been improved throughout the paper (see the answers to the questions above).

Regarding the last issue, we would be happy to present comparable results from other published methods, but we are not aware of any previous method that is aimed at this setting. What we are proposing is what we believe would be the simplest possible baseline for exploring new concepts specifically relevant for a given task using only correspondences between a visual and a word feature space. Still, we provide quantitative and qualitative evaluation that shows the correctness of the concepts discovered.

Reviewer 2 Report

The idea of ​​the work is clear, interesting and relevant. The proposed approach is logical. There are a lot of experimental studies.

However I have some suggestions:

S1 Abstract should be extended by numerical results obtained in this paper

S2 The motivation of this paper is clear. However authors should add a strong Related works section. Without it, the article looks like as a conference paper

S3 It is unclear why authors take into account only CNN. There are a lot of others more fast and interpratable neural networks architectures. They should also be analyzed in the Related works section. For example SGTM neural like structure and its modiffications (DOI: 10.1007/978-3-319-63754-9_25). Authors must justify their choice.

S4 It would be good to make some comparisons with existing approaches.

S5 The authors should describe the shortcomings of the proposed approach and the prospects for further research.

Author Response

The idea of ​​the work is clear, interesting and relevant. The proposed approach is logical. There are a lot of experimental studies.

However I have some suggestions:

S1 Abstract should be extended by numerical results obtained in this paper

We have extended the current abstract in order to convey the main conclusions of the paper. 

S2 The motivation of this paper is clear. However authors should add a strong Related works section. Without it, the article looks like as a conference paper

We have extended the introduction of our paper in order to discuss related studies on concept discovery. 

S3 It is unclear why authors take into account only CNN. There are a lot of others more fast and interpratable neural networks architectures. They should also be analyzed in the Related works section. For example SGTM neural like structure and its modiffications (DOI: 10.1007/978-3-319-63754-9_25). Authors must justify their choice.

We have improved our justification of the use of a CNN as a feature extractor in Section 2.1.

S4 It would be good to make some comparisons with existing approaches.

We would be happy to present comparable results from other published methods, but we are not aware of any previous method that is aimed at this setting. What we are proposing is what we believe the simplest possible baseline for exploring new concepts specifically relevant for a given task using only correspondences between a visual and a word feature space. We have extended the discussion on the connections between our method and other methods used for concept discovery (which are not directly comparable because they work on a different set of assumptions, such as the presence of humans in the loop or image-level concept annotations).

S5 The authors should describe the shortcomings of the proposed approach and the prospects for further research.

We address several limitations of the proposed approach in the new Section 4.4, and we have added a line about future work to the Conclusion.

Reviewer 3 Report

Authors address the interesting relationship between beauty (scenicness for landscapes) and machine learning representations (CNN as input feature to detect related concepts).

The paper is well written and clear in the organization and exposition. Unfortunately, I do not agree with the main hypothesis behind the paper (beauty can be explained by semantic concepts) and in fact I strongly disagree with any suggested cause-effect relationship (a landscape is more beautiful because it contains a mountain?). I wish reality could be so easy to model and analyze, but aesthetics is also strongly dependent on technical and cultural aspects. For example, rule of thirds, contrast, golden hour, etc.

I recommend that the authors make clear the limitations of any of these approaches. In fact what the paper really shows is how the vocabulary associated to landscape photography can be extended and this is the part I like, not the relationship with aesthetics since many variables are not studied nor controlled in the experiments.

Some specific suggestions to improve the paper (most of them are related to provide more information that although included in the refs it will be helpful to include also here):

1.Provide details about the input representation (the CNN you use) and dimensions in section 1 and 2. Your images are very helpful, so please, do the same with the text and include some specific example.

2.Extend the explanation of the datasets (sections 3) ; for example how reliable was the scoring process and if any study between the score and some metadata such as focal length or image measurements such as contrast was carried out.

3.Any photographer will tell you that in landscape photography composition is the key (specially in this low quality version of the images where focus or depth of field are not so easy to evaluate by the human eye). So I am surprised that most papers devoted to this field only show square version of the images (I understand it helps to include more images in the figure), but please, show a full image example at least; otherwise is like talking about the aesthetic use of color in movies such as The orange clockwork but printing the essay in a journal in black and white.

Author Response

Authors address the interesting relationship between beauty (scenicness for landscapes) and machine learning representations (CNN as input feature to detect related concepts).

1. The paper is well written and clear in the organization and exposition. Unfortunately, I do not agree with the main hypothesis behind the paper (beauty can be explained by semantic concepts) and in fact I strongly disagree with any suggested cause-effect relationship (a landscape is more beautiful because it contains a mountain?). I wish reality could be so easy to model and analyze, but aesthetics is also strongly dependent on technical and cultural aspects. For example, rule of thirds, contrast, golden hour, etc.

We would like to start thanking the reviewer for the comments and the time invested in them. We appreciate this comment and agree with the reviewer that details of how the photo has been taken will impact the perceived beauty in the image. However, we are interested in studying the beauty intrinsic to the place depicted in the image (this will also depend on cultural factors, since trees or mountains are surely not deemed equally scenic in different cultures). We assume that the scores given by human annotators to individual images will be affected by external factors (such as the photographers choice of a particular image composition) and transient factors (such as the lightning conditions), and we are interested in filtering these out in order to estimate the scenicness associated to the place. Studying the scenicness associated to the presence of visual concepts in the images allows us to ensure that only these place-bound elements are accounted for. This argumentation has been added to the introduction.

2. I recommend that the authors make clear the limitations of any of these approaches. In fact what the paper really shows is how the vocabulary associated to landscape photography can be extended and this is the part I like, not the relationship with aesthetics since many variables are not studied nor controlled in the experiments.

The objective of our method is indeed to expand the vocabulary of concepts that are relevant to define the aesthetic value of a place. We agree with the reviewer’s observation that we do not obtain a holistic explanation of the aesthetic value found in an image, but that is not our aim. We have improved the discussion of our objectives (Introduction) to better stress this point. 

Some specific suggestions to improve the paper (most of them are related to provide more information that although included in the refs it will be helpful to include also here):

3.Provide details about the input representation (the CNN you use) and dimensions in section 1 and 4. Your images are very helpful, so please, do the same with the text and include some specific example.

We use an ImageNet pretrained ResNet50 CNN. We have added details about the architecture and the pretraining in the new version of the paper in Section 2.1.

5. Extend the explanation of the datasets (sections 3) ; for example how reliable was the scoring process and if any study between the score and some metadata such as focal length or image measurements such as contrast was carried out.

We have improved the description of the dataset, and we now refer the readers to [2] for further details.

6.Any photographer will tell you that in landscape photography composition is the key (specially in this low quality version of the images where focus or depth of field are not so easy to evaluate by the human eye). So I am surprised that most papers devoted to this field only show square version of the images (I understand it helps to include more images in the figure), but please, show a full image example at least; otherwise is like talking about the aesthetic use of color in movies such as The orange clockwork but printing the essay in a journal in black and white.

We agree on the comment about the impact of photography composition on the scenicness, which we already discussed in the first comment. An example of six original images is provided in Figure 1. The square images we show in the results section are there mostly 1) to show that we detect the right concept for images and 2) to be able to compare the ScenicOrNot with the Broden dataset, which consists of square images.

Round 2

Reviewer 1 Report

The theme of the article is interesting and has a lot potential. This paper has a clear structure. As explained in my previous comments above, my basic reservation is that due to its no novelty and necessarily non-technical nature. Still, the fact that it points towards an interresting research fields, makes me lean towards its acceptance.

I recommend publication.

Reviewer 2 Report

Dear Authors,

thank you for the improvements of your paper.